# What is the impact of giant cell arteritis on patients' lives? A UK qualitative study

Jennifer Liddle,[1,2] Roisin Bartlam,[1] Christian D Mallen,[1] Sarah L Mackie,[3] James A Prior,[1] Toby Helliwell,[1] Jane C Richardson[1]

## ABSTRACT

**Objectives** Clinical management of giant cell arteritis (GCA) involves balancing the risks and burdens arising from the disease with those arising from treatment, but there is little research on the nature of those burdens. We aimed to explore the impact of giant cell arteritis (GCA) and its treatment on patients' lives.

**Methods** UK patients with GCA participated in semi-structured telephone interviews. Inductive thematic analysis was employed.

**Results** 24 participants were recruited (age: 65–92 years, time since diagnosis: 2 months to >6 years). The overarching themes from analysis were: ongoing symptoms of the disease and its treatment; and 'life-changing' impacts. The overall impact of GCA on patients' lives arose from a changing combination of symptoms, side effects, adaptations to everyday life and impacts on sense of normality. Important factors contributing to loss of normality were glucocorticoid-related treatment burdens and fear about possible future loss of vision.

**Conclusions** The impact of GCA in patients' everyday lives can be substantial, multifaceted and ongoing despite apparent control of disease activity. The findings of this study will help doctors better understand patient priorities, legitimise patients' experiences of GCA and work with patients to set realistic treatment goals and plan adaptations to their everyday lives.

## Strengths and limitations of this study

► This study provides the first qualitative analysis of the impact of GCA and its treatment on patients' everyday lives.

► Two approaches to sampling were adopted in recognition that patients associated with the charity PMRGCAuk were unlikely to be representative of all GCA patients, and we also acknowledge that patients who felt that GCA had a large impact on their lives may have been more likely to volunteer to participate.

► Interviews were conducted with a sample of people from different social backgrounds and age groups; future research might actively aim to recruit people from a range of ethnic and cultural groups.

► The data presented are based on patients' retrospective accounts of their experiences, which may change with time.

► Our methods did not allow us to verify the clinical diagnosis of GCA from participants' medical records or the results of medical tests they had undergone, but the fact that all participants were able to describe their glucocorticoid treatment provided reassurance that our sampling strategy identified individuals from the population of interest.

[1]Research Institute for Primary Care and Health Sciences, Keele University, Keele, UK
[2]Institute of Health and Society, Newcastle University, Newcastle upon Tyne, UK
[3]Leeds Institute of Rheumatic and Musculoskeletal Medicine, University of Leeds, Leeds, UK

**Correspondence to**
Dr Jennifer Liddle;
jennifer.liddle@newcastle.ac.uk

## INTRODUCTION

Giant cell arteritis (GCA) is the the most common form of primary systemic vasculitis and primarily affects older people.[1 2] The vasculitic process may result in ischaemic manifestations such as anterior ischaemic optic neuropathy or jaw claudication, which may produce groups of characteristic symptoms leading to clinical diagnosis. Presentation may, however, be non-specific with systemic features such as fever and weight loss and in this case diagnosis is facilitated by laboratory, histological and imaging tests.

The mainstay of treatment remains high-dose glucocorticoids (initially 40–60 mg of prednisone equivalent), aiming to control disease activity. The dose is subsequently tapered gradually over several years but

is increased in the face of relapse. Gluco-corticoid-related adverse effects result in considerable morbidity in this patient population.[3] Thus, clinical management of GCA involves a balance between the risks and burdens of the disease, and the risks and burdens of the treatment. Much has been written about this from the medical perspective; the sparse literature about patient priorities in GCA suggests that patient priorities for GCA include vision, control of arms and legs and personal care abilities.[4] Qualitative research to develop patient-relevant outcome measures in large-vessel vasculitis has been conducted with patients with Takayasu arteritis, a rarer form of large-vessel vasculitis that affects younger adults,[5] but the experience of older patients

with GCA remains relatively unexplored. According to Borg and Davidson, 'when it comes to understanding recovery, the trivialities of everyday life must be seen as anything but trivial'.[6] In order to understand how best to help patients recover from their illness, we aimed to examine GCA and its treatment from the patient perspective, through the lens of its impact on patients' everyday lives.

## METHODS
### Design
This was a qualitative study using semi-structured telephone interviews and inductive thematic analysis of data. The study received ethical approval from Keele University's Research Ethics Committee (Ref. ERP2254) and verbal and written consent was obtained from all participants (a completed paper consent form was posted back to the research team by the participant after each interview).

### Participants and data collection
Participants were recruited using two approaches. First, letters were sent to patients with GCA who had taken part in a cross-sectional survey, recruited through general practice medical records, and who had consented to further contact. Second, an email was circulated by the UK charity PMRGCAuk to its regional groups and members, inviting patients to contact the research team for further information. Only patients who reported that a clinician had diagnosed GCA were included. Purposive sampling categories included age, gender and time since diagnosis. JL and CDM developed a list of topics and prompts to guide interview discussions.

Following informed consent, RB conducted and audio-recorded semi-structured telephone interviews with recruited patients with GCA until the point where the research team agreed there were 'diminishing returns' from further data collection and that data saturation was sufficient.[7–9] Brief fieldnotes were written after each interview to record thoughts about emerging analytic themes and data coding.

### Analysis
All interview recordings were transcribed and pseudonyms used to preserve participant confidentiality. An inductive thematic approach was taken to analysis.[10] RB completed an initial categorising and line-by-line coding of all data from the transcripts using QSR NVivo V.10 and discussed the development of codes and themes with JL. JL then continued with additional analysis (in discussion with JR and CDM), developing codes and themes focusing specifically on the impact of symptoms and treatment on patients' everyday lives. After further discussion, JL and JR developed a visual representation of these themes (figure 1).

## RESULTS
Participants' ages ranged from 65 to 92 years and time since diagnosis ranged from 2 months to over 6 years. All participants were White British (table 1). Interviews lasted between 29 and 128 min. One transcript was excluded from analysis because the patient contacted the research team to say that her diagnosis of GCA had been changed to a diagnosis of lymphoma.

Patients did not conceptualise the impact of GCA as a set of discrete symptoms or side effects, but as an ongoing experience, changing over time, resulting from the combination and co-occurrence of multiple factors. Figure 1 is a visual representation of subthemes and codes within the data, demonstrating the psychological impacts, symptoms and side effects that patients experienced, and the aspects of their everyday lives requiring adaptation as a result. Patients were often unable to distinguish between disease-related symptoms and treatment-related side effects and did not make a distinction in terms of impact; the impact and combination of the experiences was most important to patients, regardless of the cause. Presentation of findings reflects this.

Two overarching themes from the qualitative data are presented: ongoing symptoms of the disease and its treatment and 'life-changing' impacts.

### Theme 1: ongoing symptoms of the disease and its treatment
Patients gave detailed accounts of the symptoms of their disease and its treatment that impacted on their everyday lives, ranging from extreme to minimal. The data in tables 2 and 3 illustrate the extent to which the experience of GCA and its treatment varied between patients, in both type and duration.

#### Types of symptoms
*"The effects it had on me, my body and my mind as well…"*
Fatigue, pain, sight loss, other visual symptoms, changes in mood and changes in physical appearance and sleep (table 2) were particularly reported as having implications for the extent to which patients could continue living their lives in the same way as they had done before the onset of GCA. The experience of permanent visual loss was associated with feelings of bereavement and vulnerability. Other generalised, persistent symptoms such as dizziness and loss of strength were also reported by patients.

#### Duration of impact
*"He says, 'I'm going to give you a course of treatment', he says, 'which is gonna last for some time—more like years than weeks', he says, 'but it will control it and hopefully we can stop it'."*
The experiences of patients varied widely, with some reporting that the impact of GCA and/or its treatment on their everyday lives had continued for many years after diagnosis (up to 5 years, and longer for those who developed new features such as sight loss), while others reported that the impact of GCA and/or its treatment on their lives had been relatively short-lived.

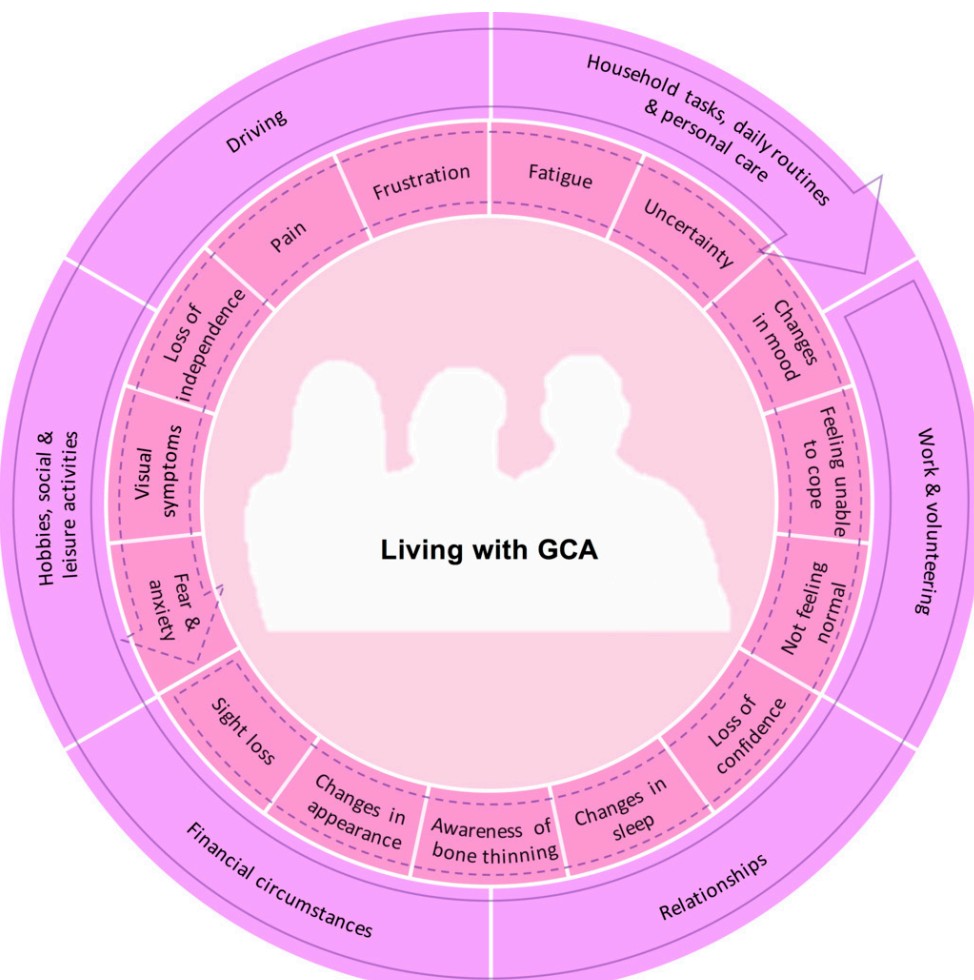

**Figure 1** This diagram represents the multiple physical and psychological symptoms experienced by patients living with a diagnosis of giant cell arteritis (GCA) (inner circle), and the multiple domains of daily living (outer circle) that these symptoms can impact on. The arrows denote the rotation of each circle, with the intention of visually representing the ability of individual/ groups of symptoms to impact on multiple domains of daily living. The rotation also indicates the potential for the relative importance and impact of each factor to vary over time due to fluctuations in the underlying disease, the dose and duration of glucocorticoid treatment, the influence of comorbidities and changes in individual adaptations and approaches to self-management.

The impact of symptoms could be categorised as: a) minimal or no long-term impact; b) continuing symptoms over time periods ranging from weeks to years or c) changes in health as a result of permanent visual loss and/or new health problems that now required management (table 3). There were no particular differences between the impacts or types of symptoms that were experienced minimally and those that became long-term.

Some patients had both long-term and permanent impacts. While some of those who experienced long-term impacts and symptoms reported that these resolved or diminished in the months and years following the initiation of treatment, others had not yet perceived such improvements. In particular, the continuation of non-specific symptoms (table 2) caused confusion and frustration for patients, especially in the context of being told that their inflammatory marker test results were within the normal range.

### Theme 2: 'life-changing' impacts
Patients whose symptoms had resolved quickly did not feel that GCA had changed their lives. However, the ongoing impacts of GCA symptoms and treatment side effects were described as being 'life changing' (Cressida, 62, 5 years 5 months since diagnosis) and 'restricting my life totally' (Dorothy, 78, 2 years 11 months since diagnosis).

#### Activities, behaviours and circumstances
*"I just used the energy on what I absolutely had to do and everything else was on hold"*
Those patients with ongoing symptoms of GCA and its treatment generally described how these had substantially affected their everyday lives including aspects of life such as: work and voluntary roles; relationships; hobbies, social and leisure activities; household tasks, daily routines and personal care; financial circumstances and driving (table 4).

Impacts could be direct (eg, as a result of fatigue or visual loss making activities unmanageable) or indirect (such as the fluctuating nature and unpredictability of symptoms affecting an individual's ability to plan or commit to future activities or events). Both types of impact required patients to adapt, consequently losing or reducing the sense of normality in their everyday lives. Patients also reflected on the feelings of guilt and regret that they had when their relationships with other people were affected.

### Thoughts and feelings
*"To be able to live a normal life, I suppose, is the thing that is desirable"*

Frustration arose as a result of patients being unable to continue with activities and tasks in the same way as they had before the onset of GCA (table 5). Patients talked about how these unwanted adaptations to their everyday lives affected the extent to which they felt independent, and the need to rely on others or ask for help was particularly unwelcome. Such reductions in independence or being unable to continue with everyday life as normal resulted in some patients feeling that they were not coping or managing the situation as well as they believed they should be able to. Their accounts demonstrated tendencies towards self-criticism rather than self-compassion, particularly when a less active, less mentally positive or less independent role was not congruent with their established perceptions of themselves and their identity.

Other psychological impacts were also prominent in patients' accounts of living with GCA. These included symptoms of low mood or depression mentioned earlier (see table 2), but reduced confidence, fear and uncertainty were also common themes in patients' accounts (table 5). Visible bodily changes resulting from glucocorticoid treatment and visual loss were two specific causes of feelings of reduced confidence and/or low mood, again impacting on perceptions of self and identity. While the actual experience of symptoms like permanent visual loss was frightening, being afraid of what might happen (ie, losing vision, experiencing a relapse of symptoms and of the potential future side effects of treatment) was a strong theme in the data. This was not exclusive to patients who had experienced ongoing or permanent symptoms.

For those who did express fears about the future, these fears were compounded by the uncertainty that patients felt about the likelihood of events such as visual loss occurring, along with their other unanswered questions about whether their symptoms would improve and which of their symptoms were due to GCA and/or its treatment or other conditions. Reassurance from clinicians sometimes helped to reduce anxiety (table 5). However, the lack of a clear-cut or predictable relationship between the length of time since the onset of symptoms and the impact of GCA and its treatment on patients' lives (table 3) was a strong feature in the data.

## DISCUSSION
GCA and its treatment present ongoing physical and psychological problems for patients, affecting their everyday lives in a wide variety of ways. As in a recent study of patients with polymyalgia rheumatica, another inflammatory condition of older people that is primarily treated with glucocorticoids, patients in our study described their experience in terms of collective impacts, rather than focusing on localised symptoms.[11] Practical, social and psychological support may be valuable therapeutic interventions to help patients recover or maintain a sense of normality in everyday life tasks, skills and activities. Patients will likely view recovery as linked with undertaking activities of value and 're-assess

| Table 1   Sociodemographic characteristics of sample (n=24) | | | |
|---|---|---|---|
| | **Male (%) n=9** | **Female (%) n=15** | **Total (%) n=24** |
| **Age group at interview (years)** | | | |
| 60–69 | 1 (11) | 4 (27) | 5 (21) |
| 70–79 | 7 (78) | 9 (60) | 16 (67) |
| 80–89 | – | 2 (13) | 2 (8) |
| 90–99 | 1 (11) | – | 1 (4) |
| **Time since diagnosis (months)** | | | |
| ≤12 | – | 4 (27) | 4 (17) |
| 13–24 | 3 (33) | 3 (20) | 6 (25) |
| 25–36 | 4 (44) | 3 (20) | 7 (29) |
| 37–48 | 1 (11) | 1 (7) | 2 (8) |
| 49–60 | – | 2 (13) | 2 (8) |
| ≥61 | 1 (11) | 2 (13) | 3 (13) |
| **Ethnicity/nationality** | | | |
| White British | 9 (100) | 15 (100) | 24 (100) |
| **Living arrangements** | | | |
| Alone | – | 5 (33) | 5 (21) |
| With one other person | 9 (100) | 10 (67) | 19 (79) |
| **Marital status** | | | |
| Married/long-term partner | 9 (100) | 10 (67) | 19 (79) |
| Single | – | 2 (13) | 2 (8) |
| Divorced/separated | – | 1 (7) | 1 (4) |
| Widowed | – | 2 (13) | 2 (8) |
| **Current work status** | | | |
| Retired | 9 (100) | 12 (80) | 21 (88) |
| Part-time work | – | 2 (13) | 2 (8) |
| Not working for health reasons | – | 1 (7) | 1 (4) |
| **Recruitment source** | | | |
| Survey | 8 (89) | 5 (33) | 13 (54) |
| PMRGCAuk charity | 1 (11) | 10 (67) | 11 (46) |

**Table 2** Ongoing symptoms reported by patients treated for giant cell arteritis

| | |
|---|---|
| Fatigue | I know **my energy levels have never been the same since** I had it. I've never been able to do what I did before. (Sue, 67, 3 years 9 months since diagnosis) |
| | I was having to lie down at meetings I felt so exhausted so intense fatigue, not just feeling a bit sleepy but intense fatigue. (June, 75, 2 years since diagnosis) |
| | You feel—you're washed out, you know? Like you're tired, very tired and washed out. That's what it feels like. (Edward, 70, 3 years since diagnosis) |
| | I'm totally exhausted all day and virtually every day. (Keith, 71, 3 years 6 months since diagnosis) |
| Pain | The only thing I've ever had, really, that is persistent, is this head pain. And it's—it's quite strange. It's almost like a weight on my head. It isn't a headache, but nobody seems to understand that. It's—it's, err, a pressure pain. [...] occasionally I get a headache but very rarely. And I can tell the difference. You know, it's just a totally different thing. (Iris, 74, 4 years 7 months since diagnosis) |
| | The only concern that I've got is that at night, I get some symptoms [...] tenderness round the ear and just on the side of the ear [...] it only ever comes at night when I'm lying down and not, not, not every night. (Neville, 74, 12–18 months since diagnosis) |
| | I've had muscle pain ever since. [...] That's 15 months after finishing taking them (steroids) I've still got muscle pain. (Christopher, 71, 2 years 8 months since diagnosis) |
| | The thing that I got that nobody else seems to have been too bothered by, was this sensitivity of the head and that was absolutely awful, because I couldn't sit anywhere, under any form of air conditioning, or in any form of draught on the shoulders or neck or head. (Barbara, 71, 2 years 10 months since diagnosis) |
| Sight loss | I've lost half—well, three quarters sight in my one eye, which was very, very traumatic. Very traumatic indeed. [...] losing your sight is like a bereavement, and I know that sounds illogical to somebody who's not experienced it, but to me it was like I'd experienced another death. I was so, so devastated [...] I just feel vulnerable. (Gloria, 72, 6 years since diagnosis) |
| | Overnight, as I said, my sight went in my left eye [...] I feel I'm fortunate; some people wouldn't think so, but I have only got part sight in my left eye. I have nothing at all in my right.(Clare, 75, 2 years since diagnosis) |
| Visual symptoms | Certainly within the last few months, **since I've been diagnosed, my vision's changed** quite a lot [...] I've noticed that I have these **blind spots**, you know, where I have no vision at all [...] For instance, if I'm looking at the television – looking at somebody's face on television, I don't see the top of their head, and that didn't happen before. (Mary, 72, 2 months since diagnosis) |
| | The side effects of this steroid are absolutely hideous [...] about 4 weeks ago, my eye—right eye started to fade and fade and fade and today, as we speak, in my right eye, I can only distinguish light and dark. I can't distinguish any shapes or people or anything but I've been to the optician and it's not blindness, as such. She says it's a, the cataract has suddenly just grown right across my eye and it's gone like that in a matter of weeks. (Sheila, 73, 11 months since diagnosis) |
| | I'd been suffering a little from blurred vision but then I'd put that down to age related and I still do have blurry vision but again, it's difficult to say, 'cos you get this sort of thing as you get older, that it was anything to do with this condition or not. (Dennis, 69, 2 years 6 months since diagnosis) |
| Awareness of bone thinning | They've also **diagnosed me now with osteo-, is it osteoporosis?** The thinning of the bones. (Laurence, 72, 1 year 7 months since diagnosis) |
| | So I went for a bone density scan and, and discovered that, you know, I was osteopenia, but everything else was alright. (Barbara, 71, 2 years 10 months since diagnosis) |

Continued

**Table 2** Continued

| Changes in mood | I'm a little bit fed up. I get **very fed up** because… just fed up with my body really (mmm) and it has debilitated my whole life now. (Sheila, 73, 11 months since diagnosis) |
|---|---|
| | I thought my life was over actually, I thought I can't go on living like this, I think I'd rather just die, it was no life at all. (June, 75, 2 years since diagnosis) |
| | I think I might have been marginally buoyed along by the steroid pills as well. […] I think also that it made me quite – I wouldn't say happy but, settled in a funny sort of way, you know? (Barbara, 71, 2 years 10 months since diagnosis) |
| | Psychologically, I suppose, you just feel down a bit and you're really fed up […] I'm afraid, under the steroids it was really nasty. I'm normally a happy, probably as you gather, chatty bunny. I became quite an intolerant bastard and wouldn't put up with anything […] I honestly believe that when they upped the dose I should have been locked up […] kept out of people's way. (Peter, 70, 6 years 8 months since diagnosis) |
| Changes in appearance | I think at first you are a bit shaken because, err, your face does get a bit bloated and I've got some photographs I won't let anyone see, err, because my face does—**I don't look like myself. My face looks too bloated.** (Rose, 73, 1 year since diagnosis) |
| | I'm huge now […] I've now put on two and a half stone […] The sweating was constant. That was awful. It was worse—far worse than the menopause ever was. Because I used to use a bath sheet to save washing the sheets – lie on a bath sheet and wear a T-shirt, cotton T-shirt, to soak the sweat up. […] and I didn't really want to go anywhere because my head was wet, you know, my hair was wet, sweating from my head. (Linda, 68, 8 months since diagnosis) |
| | The skin seems to be that thin it seems to be that it's almost like tissue so that if I do get just a small bang on it I've got a huge bruise there so all my arms are covered in bruises. (Robert, 92, 2 years since diagnosis) |
| | Just seeing my, my face – not so much my body at that point but my face morphing into a frog or a chipmunk. And then the hair down the sides of my face […] I had fat round, round the bottom of my neck […] It was just life, life changing. And my hair then went into—I've got curly hair but it went into wiry, very tight curls. It was the most peculiar thing but that was the steroids again… (Cressida, 62, 5 years 5 months since diagnosis) |
| | With the steroids […] I blew up like a balloon. […] It was the face, really. I mean I - my collar size went up about three sizes in a very short time. […] It was a damn nuisance. I had to buy a load of new shirts. […] I mean I'd got a football on my shoulders really. […] As I say, I couldn't fasten my shirt collars. (Keith, 71, 3 years 6 months since diagnosis) |
| Changes in sleep | Probably asleep by 10ish and then about half eleven, or midnight, I get—**my feet are boiling hot and throbbing like billy-o and a hideous rash.** […] In the daytime, when I'm walking around, my feet, although they look pretty hideous with a bit of a rash on and that, they don't seem to swell up and throb so much. […] **So that goes on nearly all night, every night. Consequently, I'm tired in the day.** (Sheila, 73, 11 months since diagnosis) |
| | When I was on the 60mg, I'd, I'd get an hour and half, if I was lucky, and then I'd be awake for an hour and a half and then, you know, maybe another hour; something like that, so really bad sleeping patterns right the way through. (Barbara, 71, 2 years 10 months since diagnosis) |
| | I couldn't sleep at high doses. It was a bit, yeah, wearing. But then, when I got below—when I got to 20 and below, that got a lot better. (Yvonne, 65, 4 years 7 months since diagnosis) |

Continued

| Table 2 | Continued |
|---|---|
| **Not feeling normal** | I do have, I wouldn't what I call headaches, they are more of a fuzziness, I get dizziness. But is it related to that, I don't know. (Dennis, 69, 2 years 6 months since diagnosis) |
| | I seem to have lost some strength in my arms, the strength that I did have [...] I don't seem to have full control of my legs. (Robert, 92, 2 years since diagnosis) |
| | My feet—I don't know what it is; they're very very strange [...] they just feel tight; that's the only—best way I can describe it really. [...] my feel... heavy is the word. (Clare, 75, 2 years since diagnosis) |
| | I just do feel weak somehow. It's hard to describe. (Mary, 72, 2 months since diagnosis) |
| | I really can't understand why my bloods are normal and yet [...] I still feel not normal. (Iris, 74, 4 years 7 months since diagnosis) |

continually and monitor their current well-being', comparing this with their perceptions of normality before diagnosis.[12 13]

Besides the expected impact of permanent visual loss on patients' lives,[14] it is important to note the extent to which other symptoms including fatigue, pain and 'not feeling normal' continued to affect multiple aspects of the lives of patients with GCA many years after their diagnosis. In addition, psychological consequences such as reduced confidence, anxiety and feelings of reduced independence could extend beyond the duration of the physical symptoms. Thus, the status of a patient in terms of coping, recovery and/or sense of normality in everyday life is somewhat unpredictable and does not necessarily correspond with the length of time since diagnosis/completion of treatment or clinical severity.

Ongoing adaptations were made by patients both as a direct consequence of their symptoms and as a strategy for self-management. Hall *et al* describe these strategies as part of 'the fight to maintain normality'.[12] An increased sense of independence may arise from being able to plan changes in advance. Furthermore, 'maintaining the appearance of normality'[12] can be challenging when glucocorticoid treatments cause visible bodily changes, impacting on self-image, mood and confidence, with resulting effects on activities and behaviours.

Finding the optimal balance of information provision is challenging, and by educating patients about the risks of visual loss with GCA, an additional psychological burden is introduced. GCA presents considerable risks in terms of visual loss if left untreated,[15 16] consequently clinicians may emphasise these risks to maximise patient adherence to glucocorticoid treatment. Yet fears about the possibility of visual loss can be very distressing for patients, continuing long after diagnosis. In our study, patients who had lost vision were also concerned about relapse causing further visual loss in the future. In addition, changes in vision from other conditions such as cataract and glaucoma, usually attributed to glucocorticoid therapy, heightened patients' existing anxieties about experiencing permanent visual loss as a result of GCA.

This study provides the first qualitative analysis of the impact of GCA and its treatment on patients' everyday lives. Interviews were conducted with a sample of people from different social backgrounds and age groups. The findings are compatible with existing models of health experiences and research from other chronic conditions. The data presented are based on patients' retrospective accounts of their experiences, which may change with time.

It was not the intention of this study to specifically explore ethnic or cultural factors that may impact on experiences of GCA, but future research might actively aim to recruit people from a range of ethnic groups. In addition, the impact of GCA for people of different ages and/or before and after transition points in the life-course such as retirement, and of patients who undergo treatment with evolving biologic regimens, could be

**Table 3**  Impact duration of symptoms experienced by patients with treated giant cell arteritis

| | |
|---|---|
| **Minimal or no long-term impact** | They put me **straight on to steroids. […] everything disappeared after that**. And I've had **no symptoms since.** (Joan, 81, 1 year 4 months since diagnosis) |
| | As far as living with it goes, um to be quite honest I haven't found any real change um it's not affecting me as far as my day to day living what I do and things all the way through. I think I've been very, very lucky. (David, 71, 2 years 5 months since diagnosis) |
| **Continuing symptoms** | **I'm past my year's date of my first pain** in my head and **I'm not better. I'm worse than I was at the start**…and really, just the tiredness, legs pain that I've told you […] I count these cataracts as the bane of my life. (Sheila, 73, 11 months since diagnosis) |
| | I'm pleased, you know, that I've, that I've got through it all. But that was 2010. We're now 2015. So it's only in the last, say, year that I've really got my body strong again […] it took a good few years after to really get back my strength. (Cressida, 62, 5 years 5 months since diagnosis) |
| | [I] can't do the things that I want to now […] because of the discomfort in my joints, in my arms and even in the chest. It's really, really uncomfortable […] I believe it's the reactions from the Prednisolone because I, I don't, I don't think it's from the Giant Cell Arteritis […] that seems – after about 2 month, the, the headaches went away, although I've still got the inflammation in my bloodstream like, according to (the rheumatologist), you know but… yeah it's, it's the after effects of, of those—of Pred-, Prednisolone steroids that I think is causing—this is what's caused my glaucoma problems as well. (Dennis, 69, 2 years 6 months since diagnosis) |
| **Changes in health** | You've lived your life with full vision […] It's really hard. […] because **three quarters of the sight has gone in my right eye.** (Gloria, 72, 6 years since diagnosis) |
| | I've got tablets now… I'm going to have to have long term – you see, I, I, I don't know—I'm thinking I'm going to have to have long term things to, to look after this osteoporosis now. (Laurence, 72, 1 year 7 months since diagnosis) |

explored in the future. The relationship between clinical presentation and/or comorbidities at diagnosis and the subsequent impact on life could also usefully be explored in more detail through a mixed-methods study, particularly given the clinical heterogeneity of GCA at diagnosis and the potential mediating factors such as treatment timing and approach, and individual patient behaviours and psychological responses. This could facilitate identification in advance of patients at higher risk of impacts such as anxiety and loss of normality, who might benefit most from potential support or educational interventions.

Two approaches to sampling were adopted in recognition that those GCA patients who were associated with the charity PMRGCAuk were unlikely to be representative of all patients with GCA. The sample included patients who described minimal impacts, although we acknowledge that those patients who felt that GCA had a large impact on their lives may have been more likely to volunteer to take part in the study. In our view, this does not diminish the importance of our findings about the experiences of patients who do experience ongoing impacts, despite the fact that the prevalence of these experiences has not yet been established in larger populations.

Purposive sampling is not intended to be numerically representative, but instead allows in-depth exploration and insight into experiences of patients. Consequently, the use of relative frequencies is avoided in the text to avoid confusion. Finally, our methodology did not allow us to verify the clinical diagnosis of GCA from participants' medical records or the results of medical tests they had

undergone; however, the fact that all participants were able to describe their glucocorticoid treatment provided reassurance that our sampling strategy identified individuals from the population of interest.

## CONCLUSIONS

This study demonstrates that GCA presents patients with considerable ongoing physical and psychological symptoms that affect their everyday lives in a wide variety of ways. These experiences vary over time according to the combination of multiple factors including symptoms, side effects, new health conditions and adaptations and impacts on normality in everyday life. Visible body changes attributed to glucocorticoid therapy and fear of future visual loss were important contributors to the loss of normality reported by patients.

This new understanding of the impact of GCA on patients' lives has important implications for measurement/capture of truly patient-relevant outcomes, both in clinical trials and clinical practice. It also suggests that many patients with GCA would benefit from additional psychological support, whether from peers or professionals.

Clinical management of GCA often focuses on medical concepts such as disease activity, relapse, remission and the prescribing of further medications to mitigate against the consequences of glucocorticoid toxicity.[2] We suggest that an understanding and acknowledgement of the very different ways patients experience GCA and its treatment will help clinicians negotiate patient-centred treatment plans, including planning adaptations to their lives to

**Table 4** Impacts of symptoms of giant cell arteritis and its treatment on activities, behaviours and circumstances

| Work and volunteering | I've given up a lot of my commitments. **I've had to give up a lot of my voluntary commitments**, trusteeships of this and that, and doing refereeing. (June, 75, 2 years since diagnosis) |
| --- | --- |
| | I did used to play the organ at church and various things but I can only play now to amuse myself because I can't read the music. (Clare, 75, 2 years since diagnosis) |
| | It was hard to keep working when I wasn't sleeping very well. That was probably the main thing […] I had had almost 3 months off from work […] and I'd just – I didn't want to be retired on the grounds of ill health. I love my job. I wanted to get back to it. (Yvonne, 65, 4 years 7 months since diagnosis) |
| | Having taken the steroids, you know, after my breakfast, my head was buzzing, I couldn't concentrate and I was – it was like I was having an out-of-body experience. And, and I said to my boss, 'I need to go home,' […] I sent in doctor's notes for about two or 3 months […] I don't think I could have gone back. […] there is no way on God's earth that I could have worked. I couldn't, I couldn't have done it physically […] I was sending in sick notes and […] in the end […] they just let me go. (Cressida, 62, 5 years 5 months since diagnosis) |
| **Relationships** | How my husband's stuck with me this last year, I don't know sometimes because I have – I've been, **I've been spiteful to him for no reason because I'm – don't feel quite so good and I'm ratty** […] **it's horrible and it's pathetic** and I hear myself doing it and I know I'm doing it and **I feel like really mean** for doing it to him […] and the granddaughter's been […] staying here and […] I love her so much – she's irritated me as well. […] and I said, 'Clear off. Go away' […] and it's because I'm irritable and I'm miserable. (Sheila, 73, 11 months since diagnosis) |
| | I became quite an intolerant bastard and wouldn't put up with anything and my poor old wife of 40 years had to put up with it. And I—I honestly believe that when they upped the dose I should have been locked up. I think I should have been kept out of people's way. (Peter, 70, 6 years 8 months since diagnosis) |
| | None of (my family) really knew what I was going through […] I know that my niece was a bit sceptical when I didn't got to her daughter's wedding […] And it wasn't until she saw me getting slightly better about a month, 6 weeks ago that she then realised how poorly I'd been. (Linda, 68, 8 months since diagnosis) |
| | I get a little bit short tempered than I used to be […] I react more than I used to, you know I shoot off a bit quicker. (Anne, 85, 2 years 6 months since diagnosis) |

Continued

**Table 4** Continued

| | |
|---|---|
| **Hobbies, social and leisure activities** | **I can't do my hobbies like I used to.** I used to embroider, I used to sew. I used to knit and crochet. I can't do that […] I just can't do my hobbies. And I used to enjoy cooking […] **I can't go on holiday like I—when I want to** and I can't plan anything. I don't know how I'm going to be the next day. I get up, I feel all right. We start to go out and I'm out and I'm getting bad and you know, the headaches start and my sight starts to get very deteriorated, blurred, and my joints ache. (Dorothy, 78, 2 years 11 months since diagnosis) |
| | One can't make plans. One can't say, 'Well, I'm going to go to (city) tomorrow and meet so-and-so and so-and-so,' […] You have to make a decision on the day, which is not always possible […] because you don't know how you're going to feel. (Rose, 73, 1 year since diagnosis) |
| | The steroids have made me shake. […] My whole body trembles and I, I'm—all my life, I've been a patchwork and quilter and sewer, clothes dress making and I, I can't thread a needle now […] I've had to give up my classes. […] I can't read a book. I've lost the concentration […] I'm not a person who likes to be on my own an awful lot. That's why I join clubs for sewing and grouping and photographing because I like to talk to other people and have a bit of fun and a few smiles and everything. (Sheila, 73, 11 months since diagnosis) |
| | What I have found, of course, is that I'm not covered by travel insurance, and I used to do an awful lot of travelling […] they won't insure – cover it […] I mean I like to go to Africa and places like that off the beaten track, but I'm a bit hesitant about that now. I suppose that is a big impact on my life. (Mary, 72, 2 months since diagnosis) |
| | If, for example, I go to a restaurant, I need to sit by a window so that I have the light. So it does have an impact on my life, certainly. I also find going from sunlight into a dark room very, very difficult now because I can't immediately adapt to the light changes. (Gloria, 72, 6 years since diagnosis) |
| **Household tasks, daily routines and personal care** | I struggle to cook. I struggle to do housework. **I struggle with everyday living.** (Dorothy, 78, 2 years 11 months since diagnosis) |
| | When I go shopping […] I can sort out the, the notes but it, it's the change that I have difficulty with […] My jewellery, necklaces […] I use the ones that I can put over my head that I don't have to try and fasten, but that's difficult. And I can manage to make a cup of tea and that sort of thing but, you know, it's obviously not as easy as it was, but everything takes a lot longer. […] Obviously, when you've only got part sight in the one eye, your focusing isn't right at all […] I go out in the garden now and I go to pick up some weeds and, and it takes me quite a while to get my hand on, onto the weed; I think I'm there to pick it up and I'm not […] picking it up at all. (Clare, 75, 2 years since diagnosis) |
| | Some days I, I would try and do supper, make supper but more often than not I couldn't do it. (Cressida, 62, 5 years 5 months since diagnosis) |
| | I feel very weak actually; and not a tremor but shaky. My handwriting was never brilliant but it is dreadful now. I have to type everything because it's so awful. (Mary, 72, 2 months since diagnosis) |
| | When I get up in the morning, I can't dress myself properly […] changing your underwear; I have to hang on, onto the bed, side of the bed with one hand while I negotiate the underpants and the underwear with the other hand. (Laurence, 72, 1 year 7 months since diagnosis) |
| | I was on alendronic acid for, to protect the bones but I stopped that because I found it so difficult […] even though it was only once a week […] you have to take it before anything else and not have anything to drink for at least half an hour and I found that really difficult because when I wake up I'm ever so thirsty and to have to take this pill and then wait half an hour before I could have a cup of tea I found really horrible. (June, 75, 2 years since diagnosis) |
| | I've got no osteoporosis but I do take a tablet each week for that. I have to take one a week, on the same day, every week and sit for half an hour after I've taken it. (Clare, 75, 2 years since diagnosis) |

Continued

**Table 4** Continued

| | |
|---|---|
| **Financial circumstances** | **I'm now paying for somebody to clean the house.** They come in fortnightly and have a good go through for me. (Dorothy, 78, 2 years 11 months since diagnosis) |
| | Well for one thing it's the cost really of employing people, you know, it's becomes quite expensive at times. I mean we get a female into once a week to help with housework, house cleaning, things like that. (Robert, 92, 2 years since diagnosis) |
| | At the end of the medical, he said I passed the medical, I wouldn't be entitled to any employment support allowance because, according to the medical, I had no descriptors of scored points for disability. [...] What it really meant was that none of the descriptors that they worked on to see how ill you were, applied to me. [...] they'd answer the questions on the computer and there was nothing in there for anybody with PMR and GCA. [...] I'd passed the test, but I pointed out I could only work for 90 min. (Peter, 70, 6 years 8 months since diagnosis) |
| | I went and got myself a bigger screen for my television, a bigger screen television, so I can watch the television. (Clare, 75, 2 years since diagnosis) |
| **Driving** | Well, I'm, **I'm more cautious actually, especially if I'm driving.** Although my—what I see, I see quite well now I've got my new glasses, but **the actual blind area** is a little…. It's not a problem, but **I have to be careful with traffic lights** for instance, because I can be going along and for one brief second the traffic—the red light isn't there. Then if I change just slightly my range, or look just slightly to the left or up or down, then the light is there and I know it's there and I keep my eye on—at that level then. (Mary, 72, 2 months since diagnosis) |
| | I do drive locally and some long distance but, you know, I, I don't do anything for a long period of time. That's as a result of the illness really, I think. (Barbara, 71, 2 years 10 months since diagnosis) |
| | It took me a long time to learn to go back to driving and I still won't drive on the motorway because it's my right eye that's affected and I'm frightened that I shan't see cars if I pull out. So I'm very careful. (Gloria, 72, 6 years since diagnosis) |
| | The only thing that I really miss is being able to drive…so I have to—obviously have to rely on friends and family. (Clare, 75, 2 years since diagnosis) |

**Table 5** Thoughts and feelings in everyday life with giant cell arteritis

| | |
|---|---|
| **Frustration** | It is **frustrating**: there's no doubt about it. I can't say it isn't. (Clare, 75, 2 years since diagnosis) |
| | It's very frustrating, because you—your mind still thinks you can do things, but actually […] when I try, my body flags out very quickly. (Sue, 67, 3 years 9 months since diagnosis) |
| | I get frustrated because […] I think, 'This has got to be done.' Dinners got to start and if I'm not well I say, 'Can you give me …?' 'In a minute. In a minute.' I'm not that type of person. I want to get up and get it done and sorted. And I'm getting very frustrated that way. (Dorothy, 78, 2 years 11 months since diagnosis) |
| | Oh frustrated, I suppose, really, you know. I, I want to be doing things but you know very well that you can't. (Keith, 71, 3 years 6 months diagnosis) |
| **Loss of independence** | **I can't do the jobs I used to do** about the place, you know. I'm gonna have to get a gardener in to help me with the garden because it's getting overgrown and um things like that you know and jobs around the house where I would do it myself **I've got to get someone in to do them.** […] it's just that mainly **I quite enjoyed doing it myself and I know I can't now.** (Robert, 92, 2 years since diagnosis) |
| | You just can't do things for yourself anymore and that—this is what, this is what I hate more than anything. […] So it re-, it really gets you down—it get—it does get you down sometimes […] it's just that the, the things that you, that you can't do for yourself that you've always done and, and yeah, you've got to keep on asking other people to do things for you and that, to be honest with you, that is not me. (Laurence, 72, 1 year 7 months since diagnosis) |
| | You lose your independence when you can't drive […] I do miss my driving really. (Clare, 75, 2 years since diagnosis) |
| **Unable to cope/ manage** | I really do find some things a bit of an effort […] **I feel pathetic at times** and **it's just not me** to feel pathetic. […] **I feel as if I'm kind of being stupid and making a fuss.** That **I should be** getting on with things and I—and sometimes I just can't; I just haven't got the energy. (Mary, 72, 2 months since diagnosis) |
| | It's getting that I said to my husband, 'I wish I was six foot under.' […] I'm getting quite depressed with it […] It's very demoralising when you're not the type of person that gives up. […] And I find it very hard that I'm not coping like I used to. (Dorothy, 78, 2 years 11 months since diagnosis) |
| | I find the steroids really, sort of, quite hard to cope with because they're very—they have some peculiar reactions, very, very tiring. (Rose, 73, 1 year since diagnosis) |
| **Reduced confidence** | Well **body image** is something that **may seem superficial** but when you see yourself covered in lesions all over and your hair falling out and these ghastly bruises all over it's, it **doesn't make you feel good.** (June, 75, 2 years since diagnosis) |
| | I had lost confidence in myself and I just did not have the confidence. Obviously because I've put weight on as well and I hadn't been to church for a while and, you know, people are going to see the difference in me and they're all going to say, 'Oh, what's wrong? How are you?' and all this sort of thing and you just want to stay away from people 'till you get back to normal and you don't have to answer all those questions. […] Getting hot. That can affect you, your confidence because you don't want to be with people because you start to sweat and your face goes bright red (Linda, 68, 8 months since diagnosis) |
| | Out of my right eye, I can't see […] It's taken all my confidence away. I've got no confidence at all. (Sheila, 73, 11 months since diagnosis) |
| | I felt very unattractive. […] Where I was, you know, that confident, working woman who would jump in the car and go out and see all these big corporates and, you know, enjoyed life to feeling like a really old, old person. Just, just awful. Awful. I see the photos […] I just look hideous. (Cressida, 62, 5 years 5 months since diagnosis) |

Continued

**Table 5** Continued

| | |
|---|---|
| Fear and anxiety | That is something **I'm really worried** about because **I'm petrified of going blind** [...] I've had a cataract in one eye for about five or 6 years; very slow growing. [...] then, in February this year, I just thought it was getting a bit worse [...] **I went three times to the opticians** because I was worried about my sight. **The consultant assured me that I couldn't lose my sight once I was on the steroids and so I felt quite brave then** about that [...] **I'd heard so many stories** [...] they (people) say, 'Oh, my granny went blind with that,' and all this business. [...] **Then about 4 weeks ago, my eye – right eye started to fade and fade and fade and today, as we speak, in my right eye, I can only distinguish light and dark. I can't distinguish any shapes or people or anything** [...] the cataract has suddenly just grown right across my eye and it's gone like that in a matter of weeks [...] **my biggest fear is having another relapse.** (Sheila, 73, 11 months since diagnosis) |
| | I do worry about my vision, and glaucoma can be affected by GCA as I understand. Certainly within the last few months, since I've been diagnosed, my vision's changed quite a lot. (Mary, 72, 2 months since diagnosis) |
| | It was having all these ill effects and what's it doing to my insides if it's doing this visibly to the skin and the hair? That was a worry. (June, 75, 2 years since diagnosis) |
| | I'm always in edge in case I lose more sight, and I think that's understandable, frankly. [...] I'm reducing my steroids, I'm on three milligrams a day at the moment, reducing down to 2.5 over a period of many weeks. I'm concerned that when I finally come off the steroids, is it all going to come back? That is a fear I have. (Gloria, 72, 6 years since diagnosis) |
| | I found out about GCA and prednisolone and that sort of thing, and the risk of blindness and, and that – it sort of brings you down to earth a bit more [...] I'd looked after patients on steroids and their skin was so fragile and bones crumbling, and I was thinking, 'Oh no. It's all going to happen to me.' [...] It was just the fear of the side effects of steroids. (Iris, 74, 4 years 7 months since diagnosis) |
| | ...these alendronic acid tablets which I can't take now because of the problems I've had with my gums. So at the moment I'm sort of stuck between the devil and the deep blue sea, I don't want to lose my teeth but I'll have to if, I've got to the take the tablets I don't know [...] I've got to stand up for an hour after I've taken them and I've got to have an empty tummy. I've got to do it first thing in the morning and they seem, I'm a bit scared of them to be honest. (Anne, 85, 2 years 6 months since diagnosis) |
| Uncertainty | To me, one of the most frustrating things is when I'm talking to somebody medical and they just give me **the textbook quotes** [...] **in terms of how long the condition's going to last.** I mean, some doctors are still saying, 'Oh,' you know, '2 years'. Well, maybe for some people, but **there are a lot of us out here that, you know, have had it a lot longer.** (Yvonne, 65, 4 years 7 months since diagnosis) |
| | My only concern is have I been told how severe that this condition can be and, is it something you can expect to keep reoccurring; you know, a little bit forever or after I've got rid of this and gone through this regime that we're going now and I got down to zero steroids, is that it or is it likely to recur again? (Neville, 74, 12–18 months since diagnosis) |
| | I don't know whether it's just, is it latent or is it just cured? (Laughs) I don't know how this works. (Joan, 81, 1 year 4 months since diagnosis) |
| | Well I'm just puzzled sometimes as to [...] whether I would have been like this whether I'd got this other thing, or whether, you know, it's all part of it. (Robert, 92, 2 years since diagnosis) |

help them maintain independence and retain or restore a sense of normality.

**Acknowledgements** The authors thank all the respondents who gave their time to be interviewed, and the PMRGCAuk charity for help with recruitment advertising.

**Contributors** Study design and conception: CDM, JL, JCR, TH and JAP. Data collection: RB. Data analysis and interpretation: JL, RB, JCR, SLM and CDM. Manuscript preparation: JL, RB, JCR, CDM, SLM, JAP and TH.

**Funding** This paper presents independent research funded by the National Institute for Health Research School for Primary Care Research (NIHR SPCR), Grant Reference Number 294. CDM is funded by the NIHR Collaborations for Leadership in Applied Health Research and Care West Midlands, the NIHR School for Primary Care Research and an NIHR Research Professorship in General Practice (NIHR-RP-2014-04-026). SLM is funded by an NIHR Clinician Scientist Award. The views expressed are those of the author(s) and not necessarily those of the NHS, the NIHR or the Department of Health.

**Competing interests** We have read and understood BMJ policy on declaration of interests and declare the following interests: CDM and RB had support from the National Institute for Health Research School for Primary Care Research (NIHR SPCR) for the submitted work. SLM received an honorarium for serving on a Medical Advisory Board for Chugai Roche Pharmaceuticals. All the authors have no other financial relationships with any organisation that might have an interest in the submitted work in the previous 3 years; no other relationships or activities that could appear to have influenced the submitted work.

**Patient consent** Detail has been removed from these case descriptions to ensure anonymity. The editors and reviewers have seen the detailed information available and are satisfied that the information backs up the case the authors are making.

**Ethics approval** Keele University's Research Ethics Committee.

**Provenance and peer review** Not commissioned; externally peer reviewed.

**Data sharing statement** Keele Research Institute for Primary Care and Health Sciences has established data sharing arrangements to support joint publications and other research collaborations. Applications for access to anonymised data from our research databases are reviewed by the Centre's Data Custodian and Academic Proposal (DCAP) Committee and a decision regarding access to the data is made subject to the ethical approval first provided for the study and to new analysis being proposed. Further information on our data sharing procedures can be found on the Institute's website (http://www.keele.ac.uk/pchs/publications/datasharingresources/) or by emailing primarycare.datasharing@keele.ac.uk.

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
