## [Reviewer comments · BMJ Open]

ARTICLE DETAILS

TITLE (PROVISIONAL)	What is the impact of giant cell arteritis on patients' lives?: a UK qualitative study
AUTHORS	Liddle, Jennifer; Bartlam, Roisin; Mallen, Christian; Mackie, Sarah; Prior, James A.; Helliwell, Toby; Richardson, Jane

VERSION 1 - REVIEW

REVIEWER	Matthew Koster Mayo Clinic, Rochester, MN
REVIEW RETURNED	01-May-2017

GENERAL COMMENTS	In the article entitled "The impact of giant cell arteritis on patient's lives: a qualitative study," Dr. Liddle and colleagues address an important area in the treatment of vasculitis, particularly the burden of disease and treatment from the patient perspective. The concept is original and the manuscript is overall well written. The findings regarding the fear of possible vision loss and glucocorticoid-related treatment burdens are novel and will contribute understanding to this disease process. A limitation (appropriately noted in the text) is that patients were not able to make distinction between disease-related symptoms and treatment-related side-effects. While this would be very helpful to know it is understandably difficult to qualify/quantify. Repeating such interviews or contrasting future interviews in patients who received the evolving biologic regimens will be of some potential benefit. An area that would have been helpful, but may not have been possible due to the identification of patients and thus outside the scope of this study, is understanding the patient's additional co-morbidities prior to the diagnosis and treatment of GCA and its impact on the disease/treatment course. What is relevant to the clinician in the office is trying to identify ahead of time which patients will be at higher risk for development of disease/treatment related anxiety, etc. and whether potential interventions like support groups or educational sessions would be of benefit. This could be a future area of study.
---

REVIEWER	Dr Hubert de Boysson Department of Internal Medicine, Caen University Hospital, France
REVIEW RETURNED	02-May-2017

GENERAL COMMENTS	The authors of this article aimed to describe a little known aspect of GCA regarding the impact of the disease on patient's lives.
--

	It is an interesting and well-written article. However, few clarifications are required to emphasize the impact of the results.  - My main concern regards the lack of description of included patients. GCA is a polymorphic disease with different presentations and different severities (from bilateral blindness or stroke to isolated fever without symptoms). Considering this important clinical heterogeneity at diagnosis, the subsequent impact on life should be very different. As stated by authors, patients who responded to the survey might suffer from more severe disease. - We regret not to know how many patients were asked to participate. Only 24 responded. If the 24 patients represent only a small minority of patients who were asked to participate, the impact of these results is downplayed and is clearly not representative of GCA patients. - A stratification of patients should also probably done according to the age. The impact on life of any diseases is probably different at 65 or 92 yo. - Authors should more clearly highlight the differences of disease's impact according to the GCA duration. Patients were recruited at 2 months to >6 years post diagnosis. Patients still treated after 2 years post-diagnosis might have resistant disease and subsequently experience more impact on their lives. Conversely, at 2 months, patients are more likely in remission and experience few symptoms or side effects. - I am surprised that "patients did not make a distinction between disease-related symptoms and treatment-related side-effects". It is clearly a bias of this study. GCA patients with a regular follow-up usually know their disease and are able to distinguish disease activity from treatments' side effects. - How written consent was obtained through a phone call ?
--	---

VERSION 1 – AUTHOR RESPONSE

Reviewer: 1

Reviewer Name: Matthew Koster

Institution and Country: Mayo Clinic, Rochester, MN

Please state any competing interests: None declared

R1: In the article entitled "The impact of giant cell arteritis on patient's lives: a qualitative study," Dr. Liddle and colleagues address an important area in the treatment of vasculitis, particularly the burden of disease and treatment from the patient perspective. The concept is original and the manuscript is overall well written. The findings regarding the fear of possible vision loss and glucocorticoid-related treatment burdens are novel and will contribute understanding to this disease process.

Authors: Thank you for your comments and positive feedback about the importance of understanding the patient perspective and the novel contribution of our findings for the treatment of GCA.

R1: A limitation (appropriately noted in the text) is that patients were not able to make distinction between disease-related symptoms and treatment-related side-effects. While this would be very helpful to know it is understandably difficult to qualify/quantify. Repeating such interviews or contrasting future interviews in patients who received the evolving biologic regimens will be of some

potential benefit.

Authors: We agree with these comments, and we have added the suggestion for future research to the manuscript.

R1: An area that would have been helpful, but may not have been possible due to the identification of patients and thus outside the scope of this study, is understanding the patient's additional co-morbidities prior to the diagnosis and treatment of GCA and its impact on the disease/treatment course. What is relevant to the clinician in the office is trying to identify ahead of time which patients will be at higher risk for development of disease/treatment related anxiety, etc. and whether potential interventions like support groups or educational sessions would be of benefit. This could be a future area of study.

Authors: We agree this is an important point, and we have added this as a suggested area of future research that could facilitate clinical decision making.

Reviewer: 2

Reviewer Name: Dr Hubert de Boysson

Institution and Country: Department of Internal Medicine, Caen University Hospital, France

Please state any competing interests: None declared

R2: The authors of this article aimed to describe a little known aspect of GCA regarding the impact of the disease on patient's lives.

It is an interesting and well-written article. However, few clarifications are required to emphasize the impact of the results.

Authors: Thank you for expressing your interest in our article, and for your suggestions for improvement.

R2: - My main concern regards the lack of description of included patients. GCA is a polymorphic disease with different presentations and different severities (from bilateral blindness or stroke to isolated fever without symptoms). Considering this important clinical heterogeneity at diagnosis, the subsequent impact on life should be very different. As stated by authors, patients who responded to the survey might suffer from more severe disease.

Authors: We agree that this is a limitation of our qualitative study that could be addressed by a mixed-methods approach combining a structured quantitative approach to data collection (survey or accessing medical records) with sampling for qualitative interviews according to clinical heterogeneity at diagnosis. Unfortunately this was beyond the scope of our smaller-scale qualitative study, though our findings suggest that the experience of patients is unpredictable and does not necessarily correspond with clinical severity. We have inserted a suggestion in the manuscript regarding future research to explore patients' experiences and clinical heterogeneity.

R2: - We regret not to know how many patients were asked to participate. Only 24 responded. If the 24 patients represent only a small minority of patients who were asked to participate, the impact of these results is downplayed and is clearly not representative of GCA patients.

Authors: Our sampling strategy meant that we did not control the number of patients that received an invitation to participate in the study, as emails were circulated widely by the PMRGCAuk charity. In line with theoretical approaches to qualitative research more generally, we were not aiming for a sample that was representative of all GCA patients, but we did use a purposive strategy to ensure

that we included patients with a range of characteristics within our sampling categories (age, gender, time since diagnosis). We have not used relative frequencies in the text to avoid the implication that our sample was numerically representative, and we hope that our acknowledgement of this in the strengths and limitations section of the manuscript clarifies our approach.

R2: - A stratification of patients should also probably done according to the age. The impact on life of any diseases is probably different at 65 or 92 yo.

Authors: We agree that this would be an interesting area of study, as the impact of GCA may differ across the lifecourse. We did not find any patterns in our data that indicated age was central to the experienced impact of GCA, but we have inserted a suggestion in the manuscript for future research to explore this hypothesis further.

R2: - Authors should more clearly highlight the differences of disease's impact according to the GCA duration. Patients were recruited at 2 months to >6 years post diagnosis. Patients still treated after 2 years post-diagnosis might have resistant disease and subsequently experience more impact on their lives. Conversely, at 2 months, patients are more likely in remission and experience few symptoms or side effects.

Authors: This is an interesting point that we anticipated might feature in our data. However, the status of patients in terms of coping, recovery and/or sense of normality in everyday life did not necessarily correspond with time since diagnosis and/or completion of treatment. While we did find that patients whose symptoms had resolved quickly did not feel that GCA had changed their lives much, one of our key findings was that ongoing impacts for other patients comprised more than clinical 'symptoms' alone, particularly psychological consequences and sense of normality. These impacts were often not linked with duration of the disease and/or treatment as far as we can ascertain from our data. For this reason, we have focused on duration of impact in our manuscript, rather than duration of disease/treatment, as we feel this best represents patients experiences in everyday life.

R1: - I am surprised that "patients did not make a distinction between disease-related symptoms and treatment-related side-effects". it is clearly a bias of this study. GCA patients with a regular follow-up usually know their disease and are able to distinguish disease activity from treatments' side effects.

Authors: We have amended the manuscript to clarify this point in the text. It is often difficult to clinically distinguish between symptoms and side-effects such as fatigue, visual symptoms, changes in mood, changes in sleep and not feeling normal, so we believe that it is understandable that patients also found this difficult.

In a few cases, the symptoms described (e.g. table 2) were attributed to the disease and/or its treatment by the patients we interviewed, but we were a) unable to ascertain the accuracy of these perceptions and b) attribution to disease or treatment was unrelated to the impacts described of living with GCA. Any perceived cause of a (group of) symptom(s) was unconnected to its consequent impacts on everyday life. To patients, GCA was not a set of discrete symptoms or side-effects, but an ongoing experience (changing over time) resulting from the combination and co-occurrence of multiple factors.

R2: - How written consent was obtain through a phone call ?

Authors: Thank you for noting that this explanation was missing. Participants were sent two copies of the consent form ahead of their interviews. The researcher (RB) read through each of the statements on the consent form and sought verbal consent from participants at the start of the interview.

Interviewees were asked to complete and sign the paper copy at the same time, which was then returned by post. We have added a short explanation of this in the text.